# Eigenstate control of plasmon wavepackets with electron-channel blockade

**Shintaro Takada** [1,2,3,4] ✉, **Giorgos Georgiou** [5], **Junliang Wang** [6], **Yuma Okazaki** [1], **Shuji Nakamura**[1], **David Pomaranski**[7], **Arne Ludwig** [8], **Andreas D. Wieck**[8], **Michihisa Yamamoto** [7,9], **Christopher Bäuerle** [6] & **Nobu-Hisa Kaneko** [1]

Coherent manipulation of plasmon wavepackets in solid-state systems is crucial for advancing nanoscale electronic devices, offering a unique platform for quantum information processing based on propagating quantum bits. Controlling the eigenstate of plasmon wavepackets is essential, as it determines their propagation speed and hence the number of quantum operations that can be performed during their flight time through a quantum system. When plasmon wavepackets are generated by short voltage pulses and transmitted through nanoscale devices, they distribute among multiple electron conduction channels via Coulomb interactions, a phenomenon known as charge fractionalisation. This spreading complicates plasmon manipulation in quantum circuits and makes precise control of the eigenstates of plasmon wavepackets challenging. Using a cavity, we demonstrate the ability to isolate and select electron conduction channels contributing to plasmon excitation, thus enabling precise control of plasmon eigenstates. Specifically, we observe an electron-channel blockade effect, where charge fractionalisation into cavity-confined channels is suppressed due to the plasmon's narrow energy distribution, enabling more stable and predictable plasmonic circuits. This technique provides a versatile tool for designing plasmonic circuits, offering the ability to tailor plasmon speed through local parameters, minimise unwanted plasmon excitation in adjacent circuits, and enable the precise selection of electron-channel plasmon eigenstates in quantum interferometers.

Propagating single electron wavepackets in the form of plasmonic pulses shows promising potential of quantum coherent nanoelectronics for quantum information and sensing applications[1–4]. In analogy to quantum optics, quantum information can be encoded in a flying electron, which can be controlled in flight as it propagates through well-defined electronic waveguides[1,5]. In particular, they pave the way for a novel quantum architecture featuring flying electron qubits with a reduced hardware footprint, where many qubits share

[1]National Institute of Advanced Industrial Science and Technology (AIST), National Metrology Institute of Japan (NMIJ), Tsukuba, Ibaraki, Japan. [2]Department of Physics, Graduate School of Science, University of Osaka, Toyonaka, Japan. [3]Institute for Open and Transdisciplinary Research Initiatives, University of Osaka, Suita, Japan. [4]Center for Quantum Information and Quantum Biology (QIQB), University of Osaka, Osaka, Japan. [5]James Watt School of Engineering, Electronics and Nanoscale Engineering, University of Glasgow, Glasgow, UK. [6]Université Grenoble Alpes, CNRS, Grenoble INP, Institut Néel, Grenoble, France. [7]Quantum-Phase Electronics Center and Department of Applied Physics, The University of Tokyo, Bunkyo-ku, Tokyo, Japan. [8]Lehrstuhl für Angewandte Festkörperphysik, Bochum, Germany. [9]Center for Emergent Matter Science, RIKEN, Wako, Saitama, Japan. ✉e-mail: takada@phys.sci.osaka-u.ac.jp

the same device, and enhanced connectivity[1,6]. They also mark a crucial step towards achieving quantum sensing with picosecond time resolution, addressing the need for on-chip characterisation of the quantum electromagnetic environment in ultrafast solid-state devices[4,7].

Single-electron excitations for flying electron qubit applications can be generated on demand by applying Lorentzian voltage pulses to a two-dimensional electron gas[8–10]. These electron wavepackets are collective excitations on top of the Fermi sea, and their behaviour is that of propagating plasmons[11,12]. They have attracted significant interest due to their simplicity and resilience to decoherence[13]. The purity of these excitations has been verified by methods such as tomographic reconstruction[14,15] and time-resolved measurements[16]. Building on this, these single-electron excitations have enabled groundbreaking experiments, including the observation of electron antibunching in quantum devices, driven by both Fermionic exchange statistics[10,17] and Coulomb repulsion forces[18–20]. More recently, coherent control of these excitations was demonstrated in Fabry-Pérot[4] and Mach-Zehnder interferometers[3,21]. These advances are laying the foundation for a robust platform in electron quantum optics, using ballistic plasmon wavepackets with promising potential for quantum information processing[1].

Eigenstates of a wavepacket in a quasi-one-dimensional quantum wire can be described by the bosonisation formalism[22], generalising Tomonaga-Luttinger liquid to a system containing an arbitrary number of electron conduction channels. In a quantum wire with $N$ electron conduction channels below the Fermi energy ($2N$ including spin), the Coulomb interaction creates $N$ charge modes and $N$ spin modes. Here, we use the term channel to refer to the single-particle transverse eigenstate of the Fermi liquid and mode to refer to the collective excitations in the interacting system. When a wavepacket is excited by a voltage pulse on an Ohmic contact, it is a charge excitation and hence only charge modes are excited. Among the $N$ charge modes, one fast plasmon mode and $N-1$ slower modes are formed. The plasmon mode has the highest charge-carrying capability and is the dominant eigenstate of a wavepacket excited by a short voltage pulse.

Such an interacting system has been studied in the quantum Hall regime with one or two conduction channels[23–25] and non-trivial propagation of plasmons has been revealed, such as charge fractionalisation[26,27] and spin-charge separation[28–30]. At zero magnetic fields, controlling the eigenstates of plasmon wavepackets and thus their propagation properties can be achieved by electrostatic Schottky gates[31]. By using the gates defining the quantum wire, one can control the number of available conduction channels for the plasmon mode. Since the speed of the plasmon mode is enhanced due to the Coulomb interaction between electrons in the channels contributing to the plasmon mode, compared to the Fermi velocity of Fermi liquid electrons, controlling the number of available channels can be a useful tool for controlling the plasmon speed on-demand. We note that the analogy between the plasmon mode and surface plasmon plariton (SPP) modes in an insulator-metal-insulator (IMI) structure is discussed in Supplementary Note 1. In an ideal circuit, a plasmon propagating through a single conduction channel will have almost the same speed throughout the device, and the quantum information carried by the plasmon will be controlled by the interference of the channel. However, realising a single conduction channel at zero magnetic fields is challenging due to nanofabrication imperfections and impurities because of reduced charge-screening effects. Controlling the eigenstate of plasmons in large quantum circuits through electrostatic gates can release some of the nanofabrication constraints, enabling complex circuits with multiple components and at zero magnetic fields.

In this article, we reveal a novel phenomenon, which we call electron-channel blockade for plasmon wavepackets, that can effectively suppress additional electron conduction channels in a quasi-1D wire, therefore allowing plasmon propagation into a single channel. This can be realised by forming a cavity between two local constrictions embedded in a long quantum wire. The electron-channel blockade can be triggered by an external voltage on the local gate and can be used to control on demand the eigenstate of plasmon wavepackets. Since the eigenstate is directly related to the speed of the plasmon wavepacket, it is used to control the speed and hence can be applied for a delay line in plasmonic quantum circuits. Furthermore, the electron-channel blockade allows us to suppress excitation of plasmons in circuits, including those in nearby circuits.

## Results

### Time-resolved measurement of plasmon wavepackets

Our quantum device consists of a $100\,\mu m$-long electron waveguide, fabricated by depositing electrostatic gate electrodes on a GaAs/AlGaAs heterostructure, as schematically drawn in Fig. 1a. The waveguide length can be adjusted using two segments, labelled w1 and w2. By applying a negative gate voltage to these surface gates ($V_{w1}$, $V_{w2}$), the underlying two-dimensional electron gas (2DEG) can be depleted, allowing for the formation of either a $50\,\mu m$ waveguide (w2) or a $100\,\mu m$ waveguide when both segments (w1 + w2) are combined. Electrons are injected into the 2DEG by applying a voltage pulse $V_{in}(t)$, with variable duration, to the left Ohmic contact $O_i$, using an Arbitrary Waveform Generator (AWG, Keysight M8195A). They propagate in the form of a plasmon wavepacket through the quantum device. For detection, we utilise the right Ohmic contact, $O_o$, where the current is converted into voltage across a $10\,k\Omega$ resistor, then amplified and measured. The quantum point contacts (QPCs) at the entrances of the two waveguides, highlighted in red and blue, are used to locally control the number of transmitting electron conduction channels, while the QPC at the waveguide exit, highlighted in purple, is used to measure the speed of plasmon wavepackets via time-resolved measurements. Figure 1b illustrates the shape of the voltage pulses used in the measurement of this device. For our measurements, we applied voltage pulses with temporal widths varying from 52 ps to 500 ps. The displayed pulse shapes were recorded at the output of the AWG using a sampling oscilloscope at room temperature.

Time-resolved measurements are performed using QPC3, which plays the role of an ultrafast on/off switch, thus enabling in-situ stroboscopic probing[31]. Initially, QPC3 is closed by applying a sufficiently large negative gate voltage to the DC port of the bias-tee, ensuring zero conductance across the waveguide. To probe the propagating wavepacket, QPC3 is briefly switched on by applying a 52 ps-long positive voltage pulse, $V_{det}(t)$, to the upper gate of QPC3 through the RF port of the bias-tee. This pulse allows a small fraction of the plasmon wavepacket to pass through QPC3. By varying the time delay between the generation of the plasmon wavepacket triggered by a voltage pulse $V_{in}(t)$ and the detection pulse $V_{det}(t)$ at QPC3, we can reconstruct the temporal profile of the wavepacket in a time-resolved manner. To obtain an absolute value for the plasmon propagation speed, we carefully calibrate the length difference of the RF lines, using the known propagation speed of the two-dimensional plasmon (see Methods). To obtain a measurable current, we repeat the procedure at a repetition rate of 250 MHz. In addition, we modulate the injected pulse on $O_i$ at 12 kHz and measure the output current at $O_o$ with a lock-in technique to enhance the signal-to-noise ratio. The measured current at $O_o$, represents, in principle, the convolution of the plasmon wavepacket propagating through the electron waveguide with the time-dependent conductance across QPC3. The group velocity of the plasmon wavepacket, $v_p$, is then calculated from the peak delay, $t_p$, and the length of the quantum wire, $L_{wire}$, using the relation $v_p = L_{wire}/t_p$. In the following, speed refers to the absolute value of the group velocity. According to the bosonisation formalism, plasmons have a linear dispersion relation and hence in principle their group velocity is equal to their phase velocity[32].

As a control experiment, we perform time-resolved measurements of the plasmon wavepacket generated by a 180 ps-long voltage

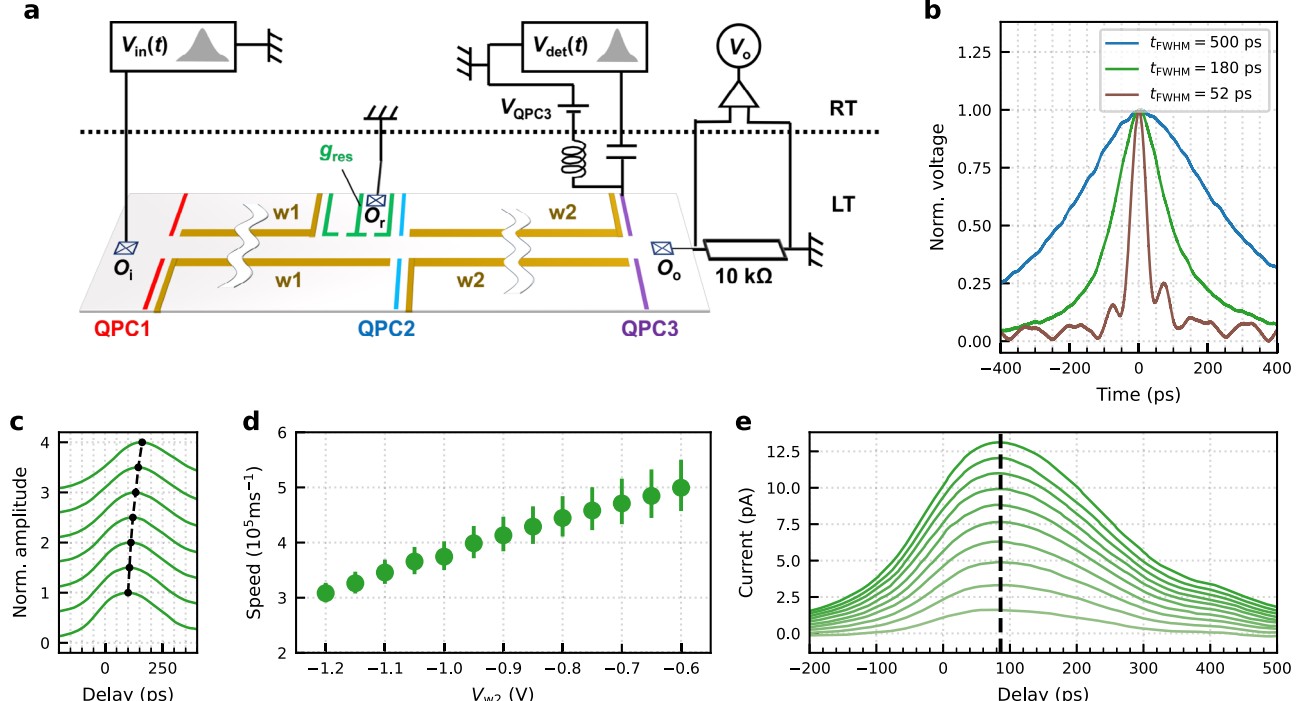

**Fig. 1 | Experimental setup and time-resolved measurement of plasmon wavepackets. a** Schematic of the device and the measurement setup. A 50 $\mu$m-long electron waveguide can be formed by polarising gates w2 and its length can be extended to 100 $\mu$m by additionally polarising gates w1, and $g_{res}$. QPC1 and QPC2 are used to locally control the number of transmitting electron conduction channels for the 100 $\mu$m and 50 $\mu$m long waveguide, respectively. QPC3 is used for time-resolved measurements of plasmon wavepackets. A high bandwidth bias tee is connected to the upper QPC3 gate to be able to apply a fast voltage pulse, $V_{det}(t)$ on top of the dc voltage, $V_{QPC3}$. The reservoir gate $g_{res}$ is used to connect the 100 $\mu$m-long electron waveguide to the Ohmic contact, $O_r$. Plasmon wavepackets are excited by applying a voltage pulse, $V_{in}(t)$, on the Ohmic contact, $O_i$. The output current at the Ohmic contact, $O_o$, is measured by the voltage, $V_o$ across a cold 10 $k\Omega$ resistor. **b** Temporal shape of the voltage pulses generated by the AWG, with pulse widths defined by full width at half maximum (FWHM) ranging from 52 ps to 500 ps. **c** Time-resolved measurement of plasmon wavepackets excited by a 180 ps-long voltage pulse, for different wire widths in the 50 $\mu$m-long quantum wire. The vertical scale is normalised to one. Each curve is offset vertically for clarity. The voltage applied on the gates w2, $V_{w2}$ is changed from −0.6 V at the bottom to −1.2 V at the top by −0.1 V step. The peak positions are indicated with black points. **d** Speed of plasmon wavepackets excited by a 180 ps-long voltage pulse as a function of the gate voltage, $V_{w2}$. The speed is calculated from the length of the quantum wire and the delay time at the peak obtained as in (**c**). **e** Time-resolved measurements of plasmon wavepackets excited by a 180 ps-long voltage pulse with varying pulse amplitudes. The peak voltage at $O_i$ is adjusted between 0.24 mV and 2.4 mV. The dashed line highlights the unchanged peak position despite varying pulse amplitudes. These characterisation measurements were conducted at 4 K.

pulse in the 50 $\mu$m-long electron waveguide, varying the side gate voltage $V_{w2}$. This reduces the number of available electron conduction channels inside the waveguide, leading to a slowing down of the propagation, as demonstrated in ref. 31. Our experimental results (Fig. 1c, d) reproduce this behaviour, and the obtained propagation speeds are consistent with the previous study[31]. In the following measurements, we employ $V_{in}$ with different temporal widths having a similar peak amplitude. The peak amplitude is slightly varied due to the bandwidth of the AWG, from 1.9 mV for the shortest (52 ps-long) voltage pulse to 2.6 mV for the longest (500 ps-long) voltage pulse. We carefully verified that for a given pulse width, the plasmon speed, or equivalently the peak position, is independent with respect to the peak amplitude. In Fig. 1e, we present representative data for a voltage pulse with a temporal width of 180 ps, demonstrating that the peak position remains constant regardless of the voltage pulse's peak amplitude

## Local control of transmitting electron channels

We now perform time-resolved measurements while controlling the number of transmitting electron conduction channels at QPC1 in the 100 $\mu$m-long electron waveguide. The voltage applied to the gates w1, w2 is set so that the waveguide accommodates more than 20 conduction channels. Figure 2a shows the results obtained with 52 ps-long voltage pulse. The pulse peak position, $t_p$, stays fixed at 100 ps and does not shift when the number of the transmitting conduction channels through QPC1 is modified. In addition, the detected peak shape has a longer tail at larger time delays, showing additional peaks.

In particular, the second peak appears around 220 ps, which is less than 3$t_p$ and hence it does not originate from the wavepacket reflected back and forth between QPC1 and QPC3. This result indicates that the plasmon wavepacket is being redistributed to the eigenstates formed by all the conduction channels in the waveguide after passing through QPC1, locally limiting the number of the transmitting conduction channels, which bears similarities to mode-matching in optics, when coupling multimode waveguide/cavity with different eigenmodes. This process is known as charge fractionalisation[24]. Our result shows the microscopic dynamics of this process.

On the other hand, when we perform the same measurement with 500 ps-long voltage pulses, a completely different result is obtained, as shown in Fig. 2b. Here, the peak position, $t_p$, shifts to a larger delay when the number of the transmitting conduction channels is locally reduced at QPC1. The speed of the plasmon wavepackets as a function of $V_{QPC1}$ calculated from the data in Fig. 2a, b is summarised in Fig. 2c. For the shorter pulse, the speed is constant as a function of $V_{QPC1}$. This is caused by the redistribution of the wavepacket into the eigenstates of the waveguide right after QPC1. For the longer pulse, the speed is controlled by $V_{QPC1}$ and hence by the number of the local transmitting conduction channels. This result implies that the charge fractionalisation process is suppressed for a longer plasmon wavepacket. Further to that, we note that the speed of the plasmon wavepackets generally decreases for those excited by longer pulses and should approach the Fermi velocity in the DC limit (see Supplementary Fig. 4 in details).

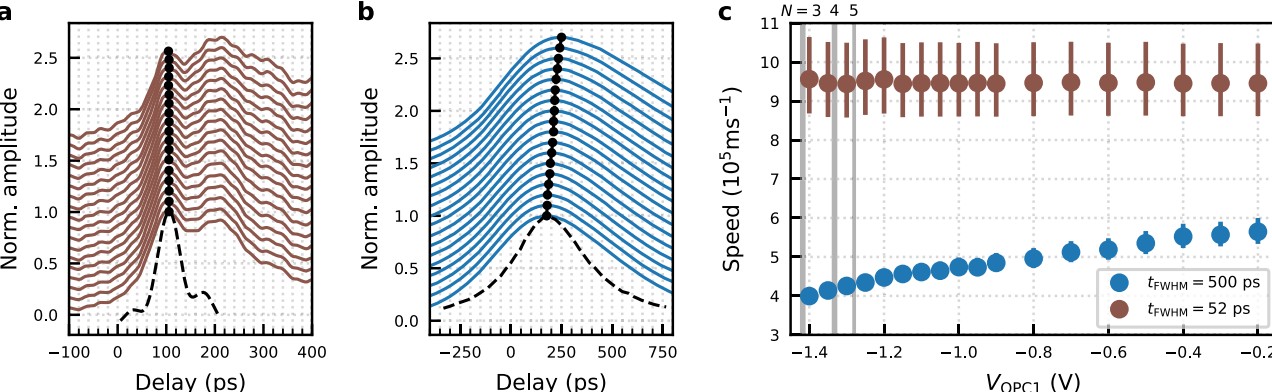

**Fig. 2 | Local control of the number of transmitting electron conduction channels in 100 $\mu$m quantum wire.** Time-resolved measurement of plasmon wavepackets excited by 52 ps-long voltage pulse (**a**) and 500 ps-long voltage pulse (**b**) for different voltages on the gates of QPC1. The amplitude is normalised to one. Each curve is offset vertically for clarity. The gate voltage $V_{\mathrm{QPC1}}$ was stepped from $-0.2$ V at the bottom to $-1.4$ V at the top. The gate voltage $V_{\mathrm{w1,w2}}$ was fixed to $-0.7$ V. The peak position is indicated by the black circles. The shape of the voltage pulse used to excite the plasmon wavepacket is drawn by the black dashed line. **c.** Speed of the plasmon wavepackets calculated from the peak delay indicated by the black circles in (**a**, **b**). Here the number $N$ on top of the grey shaded gate voltage indicates the number of transmitting electron channels across QPC1 at each voltage, which is determined by the observation of the quantised conductance.

## Fabry-Pérot cavity and electron-channel blockade

To interpret the behaviour observed in Fig. 2 we focus on the Fabry-Pérot (FP) cavity, which is formed between QPC1 at the entrance and QPC3 at the exit of the electron waveguide. For the time-resolved measurement, QPC3 is pinched off and is opened only for a short time, therefore forming one of the two end mirrors of the FP cavity. The second one is a partially transmitting end mirror to this FP cavity, formed by QPC1. As the FP cavity can be set to contain many conduction channels, when QPC1 is narrowed, electron conduction can only take place through a small number of transmitting channels. The channels that cannot transmit through the QPC are fully confined inside the FP cavity. The quantised energy levels of the FP cavity is the origin of the electron-channel blockade, as it allows for selecting the conduction channels contributing to the wavepacket transmission, when the matching conditions are met. Let us suppose that the electron waveguide contains $N$ conduction channels and QPC1 is tuned to transmit exactly 1 conduction channel. In this case, $(N-1)$ conduction channels are confined inside the cavity. The resonant cavity condition is given by $\lambda_m = 2L_{\mathrm{FP}}/m$, where $m$ is a positive integer corresponding to the mode number of the FP cavity. Similarly, the frequency components of the wavepacket that is transmitted through QPC1 should satisfy the resonant condition, $f_m = m \cdot v_{\mathrm{p}}/(2L_{\mathrm{FP}}) = m \cdot f_{\mathrm{FP}}$, where $v_{\mathrm{p}}$ is the speed of the plasmon wavepacket. On the other hand, the generated plasmon wavepacket is composed of many frequency harmonics, and its excitation spectrum has a bandwidth, $\Delta f \sim 1/t_{\mathrm{FWHM}}$, where $t_{\mathrm{FWHM}}$ is the temporal width of full width at half maximum (FWHM) of the wavepacket (see Fig. 3b, c).

When $\Delta f$ is smaller than $f_{\mathrm{FP}}$, the frequency components necessary to construct a plasmon standing wave within the cavity are not available, and the $(N-1)$ confined channels cannot contribute to the plasmon wavepacket transmission. As a result, charge fractionalisation to the $(N-1)$ conduction channels is blocked. The plasmon is funnelled and transmitted only through the eigenstate of a single electron conduction channel. We call this phenomenon as electron-channel blockade for plasmon wavepackets. In the opposite limit $\Delta f \gg f_{\mathrm{FP}}$, the Fabry-Pérot resonance condition is fulfilled, and as a result, a plasmon standing wave can be formed inside the FP cavity. For a non-interacting system, the plasmon can form a standing wave with a single conduction channel. A plasmon, however, is a collective excitation of interacting electrons. As it travels within the cavity, charge fractionalisation occurs and it will populate the eigenstates of the remaining $N-1$ conduction channels. In this situation the speed of plasmons cannot be controlled with the channel selection QPC, as the velocities within the

cavity will be renormalised due to Coulomb interactions and an $N$-channel plasmon mode appears as a main component (for a detailed theory see the Supplementary Information in ref. 31).

Figure 3 a shows $f_{\mathrm{FP}}/\Delta f$ calculated from the data in Fig. 2c for the plasmon wavepackets excited by a 52 ps and 500 ps-long voltage pulse. For the longer pulse $f_{\mathrm{FP}}$ exceeds $\Delta f$ for all $V_{\mathrm{QPC1}}$ values and hence electron-channel blockade occurs, enabling plasmon speed control with QPC1. Conversely, for the shorter pulse, $f_{\mathrm{FP}}$ is a few times smaller than $\Delta f$. In this case, the plasmon speed remains constant with respect to $V_{\mathrm{QPC1}}$. Those observations are in line with our hypothesis discussed above. For $f_{\mathrm{FP}} > \Delta f$, which translates to $L_{\mathrm{p}} = v_{\mathrm{p}} \cdot t_{\mathrm{FWHM}} > 2L_{\mathrm{FP}}$, plasmon transport through the electron conduction channels confined inside the FP cavity is blocked and hence we can control the plasmon speed by locally changing the number of the transmitting conduction channels at QPCs. For $L_{\mathrm{p}} \ll 2L_{\mathrm{FP}}$ charge fractionalisation occurs within the cavity and hence the plasmon eigenstate cannot be controlled by QPCs.

To confirm our hypothesis we intentionally break the FP cavity and demonstrate that, in this case, the QPC at the entrance of the waveguide is no longer effective to control the plasmon speed or its eigenstate. To do this, we completely depolarise the middle gate of $g_{\mathrm{res}}$, highlighted in green in Fig. 1a, to open the FP cavity towards the Ohmic contact $O_{\mathrm{r}}$. This connection to the Fermi sea reservoir breaks the FP cavity. We then perform time-resolved measurements as a function of $V_{\mathrm{QPC1}}$, for the 500 ps-long pulse, and compare the result with the configuration where the FP cavity is not broken (see Fig. 3d, e). When comparing the plasmon speed as a function of $V_{\mathrm{QPC1}}$, we observe that the plasmon travels significantly faster and shows smaller variation with $V_{\mathrm{QPC1}}$. In principle, we expect the plasmon speed to be entirely independent of $V_{\mathrm{QPC1}}$. However, this is not observed, as the opening to the reservoir accounts for less than 1% of the total length of the FP cavity. As a result, the quality factor of the cavity is reduced, but the FP cavity is still influencing the plasmon speed. To further validate our observations, we confirm that in a three-terminal device (a quantum wire with two outputs), the local selection of the number of transmitting electron channels with a QPC does not affect the speed of plasmon wavepackets, as no FP cavity is involved (see Supplementary Fig. 6.)

The demonstrated electron channel blockade using an FP cavity originates from the narrower energy spectrum of plasmon wavepackets compared to the energy quantisation of the FP cavity. This mechanism is not affected by the energy fluctuation of the reservoirs at higher temperatures. Indeed, when we perform the same

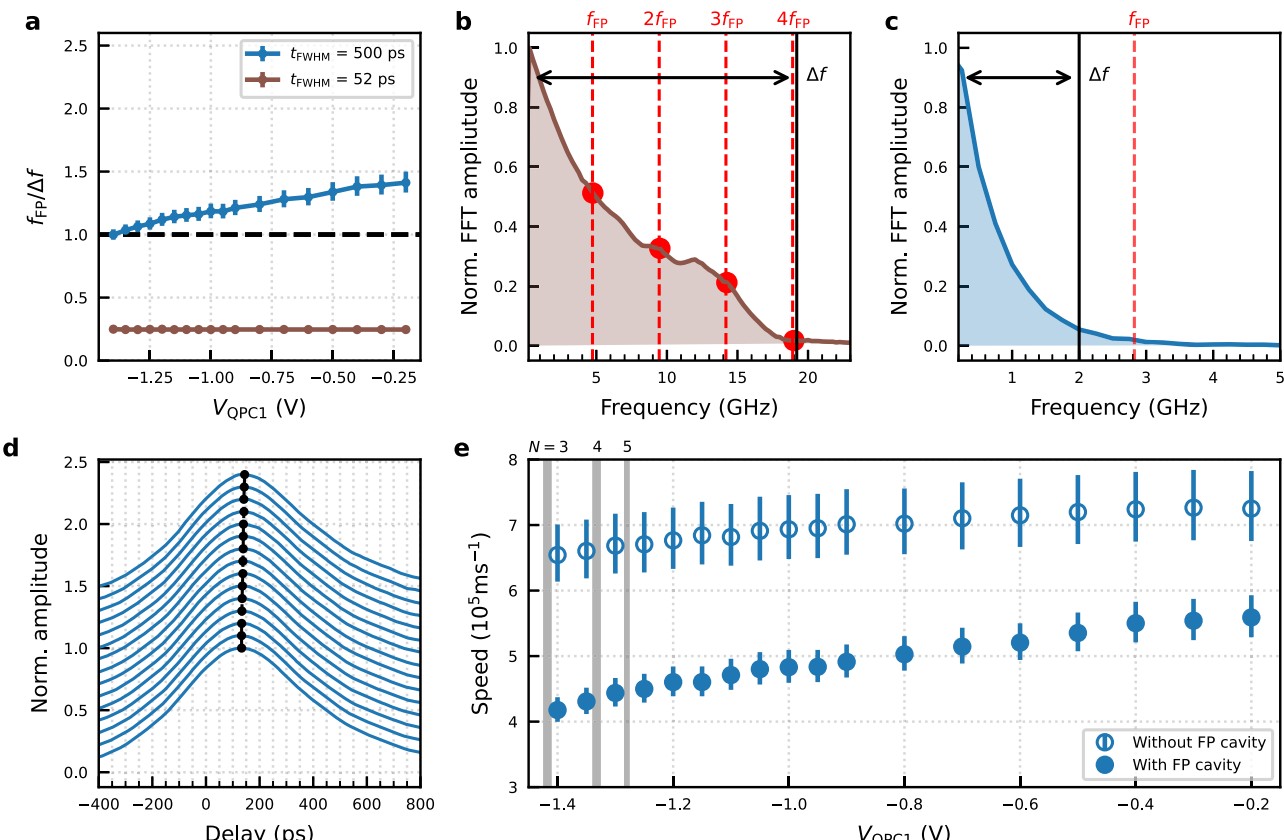

**Fig. 3 | Fabry-Pérot cavity and speed control with a local constriction. a** The ratio between the frequency quantisation of the Fabry-Pérot cavity, $f_{FP}$, and the bandwidth, $\Delta f$ for the plasmon wavepackets. The dashed line indicates $f_{FP}/\Delta f = 1$. $f_{FP}(=v_p/2L_{FP})$ as a function of $V_{QPC1}$ is calculated from $v_p$ in Fig. 2c. $\Delta f(=1/t_{FWHM})$ is calculated from $t_{FWHM}$ in Fig. 1b. Normalised amplitude of fast Fourier transform (FFT) of the voltage pulse in Fig. 1b for 52 ps-long voltage pulse (**b**) and 500 ps-long voltage pulse (**c**). The bandwidth value, $\Delta f$, is indicated by a black solid line. In addition, $f_{FP}$ at $V_{QPC1} = -0.2$ V and its multiples are indicated by the red dashed lines. **d** Time-resolved measurement of a plasmon wavepacket excited by 500 ps pulse for different gate voltages applied to QPC1 in the $100\,\mu$m-long electronic waveguide while it is connected to the Ohmic contact $O_r$. The amplitude is normalised to one. Each curve is offset vertically for clarity. The gate voltage $V_{QPC1}$ was stepped from $-0.2$ V (bottom) to $-1.4$ V (top). The peak position is indicated by the black dots. **e** Comparison of plasmon speed as a function of $V_{QPC1}$ with and without the FP cavity for the wavepackets excited by 500 ps pulse. The data without the FP cavity (new data) and the data with the FP cavity (from Fig. 2c) is provided for direct comparison. Here, the number $N$ on top of the grey shaded gate voltage indicates the number of transmitting electron channels across QPC1, which is determined by the observation of the quantised conductance.

measurement as Fig. 2 at 4 K in a slightly different gate voltage configuration, we find the same tendency (see Supplementary Fig. 5). Here, the energy fluctuation of the reservoir is much larger than $f_{FP}$ of the FP cavity. This is in clear contrast to the Coulomb blockade of electron transport through a quantum dot, where the blockade is lifted at larger energy fluctuations of the reservoirs than the quantised energy of the quantum dot at high temperatures.

**Suppression of charge fractionalisation in parallel waveguides**
The conduction of electrons in two parallel, yet electrically isolated, electron waveguides can be considered independent. However, even though direct exchange of electrons is prohibited between the two parallel waveguides, electrons are coupled through Coulomb interactions. As a result, when a plasmon wavepacket is injected into one path of the two waveguides, charge fractionalisation occurs and a plasmon wavepacket is induced at the other waveguide[23].

Here we demonstrate the suppression of the charge fractionalisation in such a parallel waveguides using the electron channel blockade demonstrated above. For this measurement, we use a device shown in Fig. 4a (see Method for details of the device). We form the two parallel electron waveguides by depleting the gates coloured in yellow. The two waveguides are electrically isolated by the middle gate along the waveguides and the upper gate at the entrance on the left, highlighted in purple. Plasmon wavepackets are injected by applying a

83 ps-long voltage pulse from an AWG (Keysight M8190A) on the left-most Ohmic contact, $O_{inj}$, with the peak amplitude of ~1.6 mV. They propagate through the lower electron waveguide and are collected in $O_l$. Here, plasmon wavepackets are expected to be induced at the upper electron waveguide[23]. To measure these induced wavepackets, we perform a time-resolved measurement using QPC$_{det}$ at the upper waveguide. For consistency, the probe pulse at QPC$_{det}$, is also 83 ps-long. In this device, we can form an FP cavity at the upper electron waveguide by using the gate, $g_{FP}$, marked in red colour. When the FP cavity is not formed, the induced plasmon wavepacket is observed as expected (the blue curve in Fig. 4b). The shape of the induced wavepacket is similar to the derivative of the injected wavepacket, whose shape is characterised in a slightly different setup from Fig. 4a (see Supplementary Fig. 7 in details). The derivative-like shape can be understood as a current induced by the capacitive coupling with the charge of the injected wavepacket. On the other hand, by applying a large negative voltage on $g_{FP}$ to deplete the 2DEG underneath and by forming the FP cavity, there is no Coulomb-induced plasmon wavepacket as shown by the orange curve in Fig. 4b. When we estimate $L_p$ and $2L$ in this situation, they are about $32\,\mu$m and $40\,\mu$m, respectively. While this does not strictly satisfy the condition for electron-channel blockade ($L_p > 2L$), the observed suppression of induced charge at $L_p \sim 2L$ strongly suggests its effectiveness. Given the limitations in precisely quantifying this condition, we attribute this suppression to

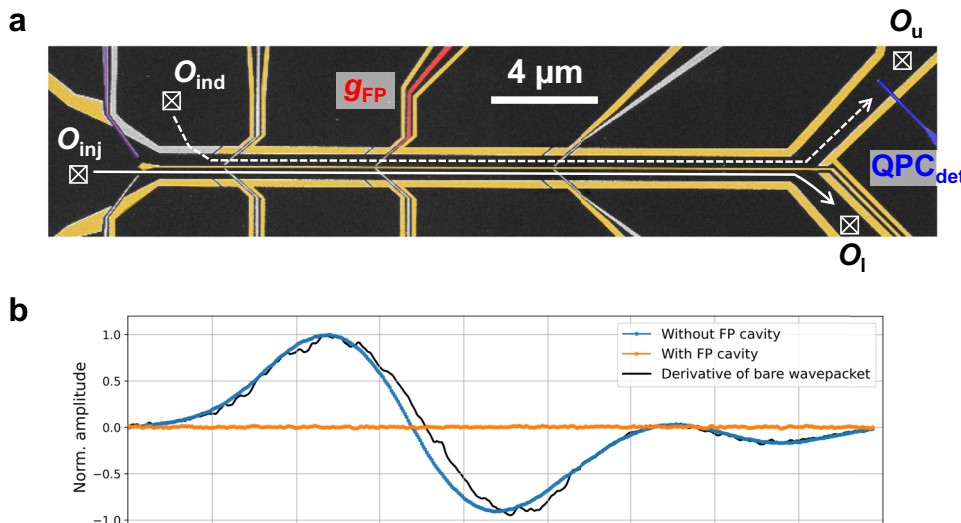

**Fig. 4 | Induced plasmon wavepacket in parallel electron waveguides. a** SEM image of the device to investigate induced plasmon wavepackets. The gates coloured in yellow and purple are used to electrostatically form the circuit for the measurement. As a result, two parallel electron waveguides, which are electrically isolated, are defined. Time-resolved measurements of the induced plasmon wavepackets, injected at the Ohmic $O_{\text{inj}}$, are performed with the gate, $\text{QPC}_{\text{det}}$. A Fabry-Pérot (FP) cavity can be formed by creating a potential barrier with the gate, $g_{\text{FP}}$, indicated in red. **b** Induced plasmon wavepacket without (blue curve) and with the FP cavity (orange curve). The amplitude is normalised with the maximum of the data without the FP cavity. The black solid curve is the derivative of the bare plasmon wavepacket measured in a slightly different setup (see Supplementary Fig. 7).

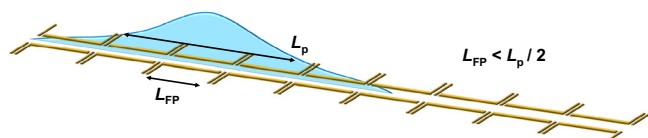

**Fig. 5 | Schematic of the proposed device structure realising a clean and long one-dimensional system.** QPCs are placed at the distance, $L_{\text{FP}}$, shorter than the half length of the plasmon wavepacket, $L_{\text{p}}/2$. When the number of transmitting channels is locally reduced to be one at each QPC, a clean and long one-dimensional system can be realised. Since the width of the quantum wires can be kept wide, the system is less vulnerable to the potential fluctuation of the surrounding environment due to the screening effect with electrons in many conduction channels.

electron-channel blockade within the FP cavity formed on the upper quantum wire.

For a more quantitative understanding of the observed behaviour, numerical simulations would be useful. In a previous work, controlling the plasmon eigenstate by modifying the width of the whole wave-guide was well understood with parameter-free numerical simulations[31]. However, direct application of the same numerical tool is not possible here due to the complex resonance behaviour of the FP cavity. The development of new numerical tools allowing for the simulation of the system with the FP cavity is desirable and will help us gain a deeper understanding of the system.

We have investigated the propagation of ultrashort plasmon wavepackets in a quantum nanoelectronic circuit while controlling the number of transmitting electron conduction channels at local constrictions. We found that an FP cavity formed by two potential barriers plays a critical role for the propagation of plasmon wavepackets. When the spatial length of the plasmon wavepacket, $L_{\text{p}}(\equiv v_{\text{p}} \cdot t_{\text{FWHM}})$, is longer than double the length of the FP cavity, electron-channel blockade occurs, where plasmon eigenstate or speed can be controlled by changing the number of the transmitting conduction channels at local constrictions. Controlling the speed of ultrashort plasmonic

excitations is attractive for quantum applications and the field of electron quantum optics. In particular, modifying the speed is equivalent to a phase delay and can be used for controlling the quantum state of a plasmon flying qubit[3,21,33]. With this method, a long one-dimensional system can be realised by placing a QPC at a distance $L$ shorter than $L_{\text{p}}/2$ as shown in Fig. 5 and setting all the QPCs to allow transmission of only a single electron conduction channel. Since most of the electron waveguide can be kept wide, the stronger screening due to higher electron density makes plasmon wavepackets less vulnerable to potential fluctuations in the surrounding environment. Furthermore, electron-channel blockade can be used to suppress unwilling leakage of plasmon wavepackets to nearby circuits. This will contribute to high-fidelity operations of a plasmon quantum state. The capability to switch on or off the induced charge or more widely the propagation of a specific plasmon eigenstate may be of interest in applications for classical plasmonic circuits, like a plasmon transistor. We expect that the demonstrated electron-channel blockade will empower precise control of plasmon wavepackets in quasi-1D electron waveguides, significantly advancing the development of both quantum and high-frequency classical circuits based on plasmons.

## Methods

### Device fabrication

The device was fabricated in a GaAs/AlGaAs heterostructure hosting a 2DEG at 110 nm (140 nm for the device in Fig. 4, Supplementary Figs. 6 and 7) below the surface. The electron density and the mobility of the 2DEG are $2.8 \times 10^{11}$ cm$^{-2}$ and $9.0 \times 10^{5}$ cm$^{2}$V$^{-1}$ s$^{-1}$ ($2.1 \times 10^{11}$ cm$^{-2}$ and $1.9 \times 10^{6}$ cm$^{2}$V$^{-1}$ s$^{-1}$ for the device in Fig. 4, Supplementary Figs. 6 and 7) at 4 K, respectively. The Schottky gates to define the quantum wires and the QPCs were defined by Ti/Au and the Ohmic contacts were defined by Ni/Ge/Au/Ni/Au alloy. A scanning electron microscope image of the device in Figs. 1–3 is shown in Supplementary Fig. 1. All measurements except for the ones in Fig. 1b, e were performed at the base temperature of a dilution refrigerator around 15 mK. Characterisation measurement in Fig. 1b was performed at room temperature and the one in Fig. 1e was performed at 4 K.

## Calibration of RF lines

The time delay between the pulses travelling through the RF injection and the detection lines is influenced by the attenuators placed on each line. The delay is fixed for each RF line and can be of the order of ps. To accurately evaluate the speed of plasmon wavepackets this time delay should be properly calibrated. For this, we used a two dimensional (2D) plasmon excitation, as in Ref. 31, since a 2D plasmon is known to be as fast as $v_{2D} = 1.0 \times 10^7$ m/s[34,35]. When we set the voltage of the gates forming the quantum wire to zero, the excited plasmon wavepackets propagate through the device as 2D plasmon excitation, which is detected at QPC3. The time-resolved measurement of the 2D plasmons excited by the voltage pulses with different temporal lengths are plotted in Supplementary Fig. 2. From the time delay at the peak of each curve, we obtain the relative time delay between the two RF lines as 112 ± 7 ps.

This time delay, $t_{2D}$, is expressed by $t_{2D} = (L_0 + 100\,\mu m)/v_{2D} + \delta_{line}$, where $L_0$ is the distance between the injection contact, $O_i$ and QPC1, $v_{2D}$ is 2D plasmon velocity, and $\delta_{line}$ is the time delay between the RF lines connected to $O_i$ and QPC3. For the time-resolved measurement the delay at the peak, $t_{peak}$ is $t_{peak} = (L_0 + 100\,\mu m - L_{wire})/v_{2D} + L_{wire}/v_p + \delta_{line}$, where $L_{wire}$ is the length of the quantum wire used as a waveguide. Therefore, $t_p = L_{wire}/v_p$ is equal to $(t_{peak} - (t_{2D} - L_{wire}/v_{2D}))$. We consider that $v_{2D} = 1.0 \pm 0.2 \times 10^7$ m/s. As a result, we use 102 ps (107 ps) as a zero time delay for the plasmon wavepacket time-resolved measurement data with the uncertainty of $\delta_{calibration} = 10$ ps (9 ps) for the data obtained in $100\,\mu m$ ($50\,\mu m$) quantum wire. The same calibration is performed for the device in Fig. 4. Then the plasmon velocity is calculated by $v_p = L_{wire}/t_p$. Here, the total uncertainty of $t_p$, $\delta_{tot} = \delta_{calibration} + \delta_{fitting}$ is used to calculate the error bar of the measurement.

## Data availability

The datasets generated and analysed during the current study are available in the Zenodo repository, [https://doi.org/10.5281/zenodo.16879800]. All other relevant data are available from the corresponding author upon request.

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

## Acknowledgements

S.T. and M.Y. acknowledge JST Moonshot (grand number JPMJMS226B). S.T., D.P. and M.Y. acknowledge Japan Society for the Promotion of Science, Grant-in-Aid for Scientific Research S (grant number JP24H00047). G.G. acknowledges EPSRC "QUANTERAN" (grant number

EP/X013456/1) and Royal Society of Edinburgh projects "TEQNO" and "TFLYQ" (Grant number 3946 and 4504). J.W. acknowledges the European Union H2020 research and innovation programme under the Marie Sklodowska-Curie grant agreement No. 754303. A.D.W. and A.L. thank the DFG via ML4Q EXC 2004/ 1-390534769, the BMBF-QR.X Project 16KISQ009 and the DFH/UFA Project CDFA-05-06. M.Y. and N-H.K. acknowledge CREST-JST (grant number JPMJCR1876). C.B. acknowledges funding from the French Agence Nationale de la Recherche (ANR), project ANR QCONTROL ANR-18-JSTQ-0001. C.B. acknowledges funding from the Agence Nationale de la Recherche under the France 2030 programme, reference ANR-22-PETQ-0012. This project has received funding from the European Union H2020 research and innovation programme under grant agreement No. 862683, "UltraFastNano". The present work has been done in the framework of the International Research Project "Flying Electron Qubits"-"IRP FLEQ" CNRS-Riken-AIST-Osaka University. Views and opinions expressed are those of the author(s) only and do not necessarily reflect those of the European Union or the granting authority. Neither the European Union nor the granting authority can be held responsible for them.

## Author contributions

S.T. conceived the experiment, performed the measurements with support from Y.O., S.N., D.P., and N-H.K., and analysed the data with input from G.G., M.Y., and C.B. G.G. and J.W. fabricated the samples. A.L. and A.D.W. provided the high-quality GaAs/AlGaAs heterostructure. S.T., G.G., M.Y., and C.B. wrote the manuscript with feedback from all authors.

## Competing interests

The authors declare no competing interests.
