## [Transparent Peer Review file · Nature Communications]

Eigenstate control of plasmon wavepackets with electron-channel blockade

Corresponding Author: Dr Shintaro Takada

Version 0:

Reviewer comments:

Reviewer #1

(Remarks to the Author)

The work reports an experiment on plasmon wavepackets in waveguides, specifically on their control by voltage pulses. A main feature is the time-resolved detection.

The manuscript represents a solid piece of work that is presumably correct (as far as I can judge from my theorist's perspective). In principle it can be published. Nevertheless, I recommend to consider the following critique.

1. I am confused by the prominent use of "eigenstate". The term is only meaningful with an associated operator, usually the Hamiltonian. However, wavepackets usually are superpositions of eigenstates. Is this the case here? Or is it adiabatic following to an instantaneous eigenstate? This should be clarified.
2. The data look surprisingly smooth. For example, the positions of the peaks (fig.1d) vary only a little on the scale of their width (fig.1c). Hence, I would expect large error bars in figure 1d. The same holds also for some panels of figures 2 and 3. A critical discussion of the precision is required.

With these items addressed, the manuscript should be suitable for publication.

Reviewer #2

(Remarks to the Author)

Review of Nature Communications Manuscript: "Eigenstate control of plasmon wave packets with electron-channel blockade" by Takada et al.

Please accept my apologies for the delayed response due to health issues and upcoming relocation.

The authors demonstrate experimentally an interesting concept of controlling the velocity, transmission and coupling of ultrashort plasmonic wavepackets in 1D quantum wire electron waveguides with controllable width, Fabry-Perot resonators, and pairs of waveguides integrated with FP resonator. The main mechanism for the control is by controlling the number of conduction channels/modes the plasmonic wavepacket occupies, which can be done by controlling the wavepacket length and the waveguide geometry. In the context of FP geometry, the authors identify an electron-channel blockade mechanism. The results are convincing and well presented. I am in favor of recommending publication with the following suggested revisions. These mostly have to do with asking the authors to better explain the physical mechanism behind how the charge fractionalization changes the dependence of the velocity over voltage. I have to admit I didn't fully understand this - maybe because I do not come from this field - but then again, it might be useful to explain it better for readers like myself. See further comments below.

1. "is used to measure the speed of plasmon wave packets via time-resolved measurements following reference [31]." - I suggest the authors give a brief description of the measurement technique.
2. V_w^2 which to my understanding controls the effective waveguide/channel width/boundary condition to change the wavepacket speed, is not shown in Fig. 1a.

3. The mechanism of charge fractionalization, at least from the authors' description in the first paragraph of Section 2, and later in section 3, seems similar to mode-matching when coupling to multimode waveguide/cavity. If correct, perhaps the authors can consider this analogy to help readers from an optics background understand it better.
4. Also, can the authors briefly explain the physical mechanism of why having too many electron channels modifies the plasmon speed? For example, it is not exactly clear to me why introducing more channels/modes cancels the change in group velocity. Perhaps the different channels/modes already have disparate group velocities such that the change in voltage causes slightly different changes in velocity of different modes, and the overall change averages out? I am not sure this is the correct explanation, but the authors should try and provide an intuitive one if possible. In addition, what happens differently in Extended Data Fig. 3 that suddenly we do see this dependence of velocity on voltage?
5. Possible typo in the caption of Fig. 3: "The data with the FP cavity are the same as in Fig. 2c" – perhaps you mean "the data without the FP cavity"?
6. In the FP cavity section: while I understand the blockade mechanism, I am still confused about how it relates to controlling the group velocity (this connects to point 4 above)
7. I really like section 4, which shows control over induced plasmonic wavepackets using the FP setup. Can the authors propose interesting devices based on this effect (e.g. transistors for plasmons)? Also, as the authors imply that the conditions for the blockade are not completely met, can they propose other possible mechanisms for the suppression (other than the electron channel blockade) that might take effect here?
8. Last paragraph typo: "quasi-1D electron waveduides"

Reviewer #3

(Remarks to the Author)

Recommendation: Major revision.

Please see the attached PDF for my full report.

[Editorial Note: This attachment is displayed at the end of the file]

Version 1:

Reviewer comments:

Reviewer #1

(Remarks to the Author)

My main critique on the original manuscript was a confusing terminology (eigenstate) and the missing discussion of the data accuracy. In their revision, the authors considered this critique. After reading the revised version, I have no further comment and recommend publication.

Reviewer #2

(Remarks to the Author)

The authors made the necessary changes in response to my review so I recommend publication

Reviewer #3

(Remarks to the Author)

I thank the authors for the detailed responses provided to the reviewer's previous comments. While I am satisfied with the clarifications on most of the concerns raised, a further point has emerged from the authors' response.

I would like to inquire whether the "plasmon eigenstate" described in the manuscript can be understood in a similar fashion to the SPP modes found in IMI (insulator-metal-insulator) structures, where SPPs split into high-energy and low-energy branches. The splitting width is modulated by the core thickness, becoming larger as the thickness decreases, resulting in a larger wave number modulation. The bosonization formalism presented in this paper appears to exhibit a similar phenomenon, where the plasmon velocity and mode characteristics are modulated by the transverse channel configuration, analogous to how the SPP splitting in IMI systems is controlled by the thickness of the intermediate layer.

The reviewer seeks confirmation whether this analogy holds. Addressing this analogy could potentially broaden the article's understanding and impact within the classical surface plasmonics community.

We would like to thank all the reviewers for his/her careful reading of our manuscript and for the insightful and constructive comments. Below, we provide a detailed point-by-point response to each of the issues raised.

Sincerely,

Shintaro Takada

(on behalf of all authors)

Reviewer #1 (Remarks to the Author):

The work reports an experiment on plasmon wavepackets in waveguides, specifically on their control by voltage pulses. A main feature is the time-resolved detection.

The manuscript represents a solid piece of work that is presumably correct (as far as I can judge from my theorist's perspective). In principle it can be published. Nevertheless, I recommend to consider the following critique.

1. I am confused by the prominent use of "eigenstate". The term is only meaningful with an associated operator, usually the Hamiltonian. However, wavepackets usually are superpositions of eigenstates. Is this the case here? Or is it adiabatic following to an instantaneous eigenstate? This should be clarified.

- We thank the reviewer for raising this important point regarding our use of the term "eigenstate". In the revised manuscript, we have clarified that "eigenstate" refers specifically to the transverse confinement eigenstates of the Hamiltonian of the interacting quasi one-dimensional system, not to single-particle states.

According to the bosonisation theory for plasmon excitations (details provided in the supplementary materials of [G. Roussely et al., *Nat. Commun.* 9, 2811 (2018)]), when there are N one-dimensional channels due to transverse confinement (or $2N$ including spin), electron-electron interactions lead to the formation of N spin modes and N charge modes. In our manuscript, we use the term "channel" to refer to the single-particle transverse eigenstates of the Fermi liquid, and "mode" to refer to the collective excitations in the interacting system, i.e., Luttinger liquid.

Since spin modes do not carry charge, they are not excited in our experiment, where charge is injected via a voltage pulse applied to the Ohmic contact. Among the N charge modes, one fast plasmon mode and $N-1$ slower modes are formed. The plasmon mode has the highest charge-carrying capacity and dominates the signal in our current detection scheme.

Therefore, when we refer to "eigenstate" in the manuscript, we mean the plasmon mode associated with a specific number of transverse channels. A "different eigenstate" corresponds to a plasmon mode formed under a different channel configuration. The wavepackets we excite propagate along this plasmon mode. Since bosonisation theory predicts a linear dispersion for plasmons, the wavepacket shape follows that of the voltage pulse and maintains its form during propagation.

We have revised the manuscript to clarify this terminology and added a more precise explanation in the relevant section [**Correction 1**].

2. The data look surprisingly smooth. For example, the positions of the peaks (fig.1d) vary only a little on the scale of their width (fig.1c). Hence, I would expect large error bars in figure 1d. The same holds also for some panels of figures 2 and 3. A critical discussion of the precision is required.

- We appreciate you bringing this to our attention. We acknowledge that the original figures showed an inconsistency between the plotted velocity values and their corresponding error bars. While we converted the velocity values from 10^6 m/s to 10^5 m/s during the analysis to improve readability on the plots (e.g., Fig. 1d, 2, and 3), we mistakenly failed to apply the same scaling factor to the error bars. The corrected error bars are now properly displayed in the revised manuscript, which should resolve the visual inconsistency you noted [**Correction 16**]. We have also added a more detailed discussion on the experimental uncertainties in the Methods section [**Correction 13**].

The primary sources of uncertainty in our experiment are as follows:

1. **Waveguide Length:** The uncertainty in the length of the waveguide, defined by electron beam lithography, is estimated to be less than 0.1 % of the total length (50 μm or 100 μm).
2. **Delay Time Measurement:** The delay time is extracted from the peak position of the signal (e.g., Fig. 1c) using a fitting procedure. The uncertainty from this fitting is less than 1 ps.
3. **RF Line Calibration:** The most significant source of uncertainty comes from the skew calibration between the RF lines (See also the supplementary materials of [G. Roussely et al., *Nat. Commun.* 9, 2811 (2018)]). We calibrated this by measuring the propagation of 2D plasmons, which have a known velocity of 1.0×10^7 m/s. Our measurement of the 2D plasmon delay time yielded a value of 112 ± 7 ps (Extended data figure 2). We also include 20 % of error on the 2D plasmon velocity ($1.0(\pm 0.2) \times 10^7$ m/s). By combining the uncertainties from both the 2D plasmon velocity and our measurement, we estimate the total uncertainty in our delay time to be approximately 10 ps.

Reviewer #2 (Remarks to the Author):

Review of Nature Communications Manuscript: “Eigenstate control of plasmon wave packets with electron-channel blockade” by Takada et al.

Please accept my apologies for the delayed response due to health issues and upcoming relocation.

The authors demonstrate experimentally an interesting concept of controlling the velocity, transmission and coupling of ultrashort plasmonic wavepackets in 1D quantum wire electron waveguides with controllable width, Fabry-Perot resonators, and pairs of waveguides integrated with FP resonator. The main mechanism for the control is by controlling the number of conduction channels/modes the plasmonic wavepacket occupies, which can be done by controlling the wavepacket length and the waveguide geometry. In the context of FP geometry, the authors identify an electron-channel blockade mechanism. The results are convincing and well presented. I am in favor of recommending publication with the following suggested revisions. These mostly have to do with asking the authors to better explain the physical mechanism behind how the charge fractionalization changes the dependence of the velocity over voltage. I have to admit I didn't fully understand this - maybe because I do not come from this field – but then again, it might be useful to explain it better for readers like myself. See further comments below.

- 1. “is used to measure the speed of plasmon wave packets via time-resolved measurements following reference [31].” – I suggest the authors give a brief description of the measurement technique.**
 - Thank you for pointing out this. Indeed, in our manuscript we provide a detailed description of the time-resolved measurements. This is outlined in the paragraph following the Reviewer's quoted text. However, we agree that this sentence is indeed misleading and broad audience may be confused. We have removed the phrase “following reference [31]” to avoid confusion in the revised manuscript [**Correction 2**].

- 2. V_w which to my understanding controls the effective waveguide/channel width/boundary condition to change the wavepacket speed, is not shown in Fig. 1a.**

- It is indeed the voltage applied on gate w2 to control the width of the 50 μm quantum wire (waveguide). In Fig. 1a we define the name of the gates and the voltage applied on the gate is expressed as $V_{\text{gate_name}}$. We have defined V_{w1} and V_{w2} when we refer to the role of w1 and w2 in the revised manuscript [**Correction 4**].

3. The mechanism of charge fractionalization, at least from the authors' description in the first paragraph of Section 2, and later in section 3, seems similar to mode-matching when coupling to multimode waveguide/cavity. If correct, perhaps the authors can consider this analogy to help readers from an optics background understand it better.

- We thank the reviewer for the insightful suggestion. Indeed, the analogy with mode-matching in optics provides a helpful perspective, especially for readers with a background in photonics or waveguide theory.

In our system, the wavepacket enters from one interacting quasi-one-dimensional region into another interacting region with a different configuration of transverse channels. This is distinct from the conventional scenario of charge fractionalization, where an electron typically enters from a non-interacting region into a strongly interacting one-dimensional system.

Nonetheless, the underlying mechanism remains similar: when the wavepacket does not match the eigenmodes of the target system, it is decomposed into multiple eigenmodes. In our case, this mismatch leads to the excitation of several charge modes, including the dominant plasmon mode and slower modes – particularly evident in the case of the shorter wavepacket (FWHM = 52 ps) in Fig. 2a. This is conceptually analogous to mode-matching in optics, where an incoming wave is decomposed into the eigenmodes of a multimode waveguide or cavity due to imperfect overlap.

We have incorporated this analogy into the revised manuscript to aid understanding for readers with an optics background [**Correction 8**].

4. Also, can the authors briefly explain the physical mechanism of why having too many electron channels modifies the plasmon speed? For example, it is not exactly clear to me why introducing more channels/modes cancels the change in group velocity. Perhaps the different channels/modes already have disparate group velocities such that the change in voltage causes slightly different changes in velocity of different

modes, and the overall change averages out? I am not sure this is the correct explanation, but the authors should try and provide an intuitive one if possible. In addition, what happens differently in Extended Data Fig. 3 that suddenly we do see this dependence of velocity on voltage?

- We thank the reviewer for this insightful question. The physical mechanism can be understood within the bosonisation formalism for interacting one-dimensional electronic systems (Luttinger liquids), which we have elaborated on in the revised manuscript [**Correction 1, 14**].

1. Physical mechanism for plasmon speed and its dependence on channel number

The reviewer's intuition that different modes have different group velocities is partially correct, but the overall effect is not a simple averaging. In a quantum wire with $2N$ single-particle channels (including spin), the Coulomb interaction creates N charge modes and N spin modes. Since our experiment uses voltage pulses to excite charge, only the N charge modes are relevant. These charge modes are not equivalent; they consist of one "plasmon mode" that carries most of the charge and is strongly affected by the Coulomb interaction, and $(N-1)$ "slow modes" that carry very little charge and are weakly affected.

The wavepackets we detect are predominantly formed by the plasmon mode. The speed of this plasmon mode is determined by the strength of the Coulomb interaction, and it is enhanced as the number of contributing single-particle channels (N) increases. In the bosonisation formalism, this plasmon mode has a linear dispersion relation, meaning its phase velocity is in principle equal to its group velocity. Therefore, our measurements of the wavepacket group velocity should directly reflect the plasmon mode's speed. We have added more explanation of this mechanism to the introduction of revised manuscript.

2. The different behaviour in Extended Data Fig. 3

The voltage dependence of the velocity observed in Extended Data Fig. 3 arises from the dynamics of the Fabry-Pérot (FP) cavity. In our setup, we control the number of channels contributing to the plasmon mode not by changing the overall wire width (as in Fig. 1d), but by locally confining them with the QPCs that form the FP cavity.

The key parameters here are the full width at half maximum (FWHM) of the plasmon wavepacket (L_p) and the length of the FP cavity (L).

- When $L_p > 2L$: The wavepacket is long, and its energy distribution is narrower than the energy quantum of the FP cavity. In this regime, the single-particle

channels confined within the FP cavity cannot contribute to the plasmon mode. Consequently, the plasmon speed is controlled by changing the confinement of the entrance QPC, as shown in Fig. 2 for the 500 ps wavepacket.

- When $L_p \ll 2L$: The wavepacket is short, and its energy distribution is much broader than the FP cavity's energy quantum. Here, the channels inside the FP cavity can contribute to the plasmon mode, and the plasmon speed becomes independent of the voltage on the entrance QPC. This is the case for the 52 ps wavepacket in the 100 μm quantum wire (Fig. 2).

In the case of Extended Data Fig. 3a, which uses a 50 μm wire, the condition $L_p \ll 2L$ is not satisfied for the shorter wavepacket (52 ps), but the wavepacket is also not long enough to fully satisfy $L_p > 2L$. This is an intermediate, or “boundary”, situation. The result is a shift in the peak delay as a function of QPC voltage, but the main peak is followed by several sub-peaks. This suggests that the wavepacket is not in a well-defined eigenstate and spreads into several different charge modes after passing through QPC2. Thus, the FP cavity dynamics are responsible for the voltage dependence seen in this particular dataset.

5. Possible typo in the caption of Fig. 3: “The data with the FP cavity are the same as in Fig. 2c” – perhaps you mean “the data without the FP cavity”?

- Thank you for pointing out this potential confusion. The reviewer's interpretation, though incorrect, highlights that our caption was indeed unclear.

In the context of the experiment, Figs. 3d and 3e show new data where the FP cavity is intentionally “broken” (i.e. a FP cavity is opened to the extra reservoir, O_r). The comparison in Fig. 3e is between the new data **without** the FP cavity and the data **with** the FP cavity from Fig. 2c.

Therefore, the phrase “The data with the FP cavity are the same as in Fig. 2c” is technically correct, but the wording is misleading. To prevent future confusion, we have revised the caption to more clearly explain the comparison being made [**Correction 10**].

6. In the FP cavity section: while I understand the blockade mechanism, I am still confused about how it relates to controlling the group velocity (this connects to point 4 above)

- This is discussed in point 4.

7. I really like section 4, which shows control over induced plasmonic wavepackets using the FP setup. Can the authors propose interesting devices based on this effect (e.g. transistors for plasmons)? Also, as the authors imply that the conditions for the blockade are not completely met, can they propose other possible mechanisms for the suppression (other than the electron channel blockade) that might take effect here?

Thank you for the kind words regarding section 4 and for the excellent questions. Your feedback highlights important avenues for future work and helps us clarify our current findings.

1. Proposed Devices Based on the Effect

We believe our work provides a foundation for several interesting new devices, with applications in both classical and quantum electronics.

- **Plasmon Transistor:** As you suggested, our method can be used to realize a plasmonic transistor. By controlling the voltage on the QPCs that form the Fabry-Pérot (FP) cavity, we can effectively switch on or off the propagation of a specific plasmon eigenstate. This control over the plasmon's eigenstate and speed allows for a "transistor-like" function where the state of the plasmon can be modulated, and its propagation can be gated.
- **On-Chip Delay Lines and Quantum Interferometers:** Our ability to precisely control the plasmon speed via the number of channels contributing to the plasmon mode is directly applicable to creating on-chip delay lines. This control is crucial for quantum interferometers, as it allows for tailoring the plasmon's flight-time and, consequently, the number of quantum operations that can be performed during propagation.
- **Minimizing Unwanted Crosstalk:** The electron-channel blockade also allows for suppressing unwanted plasmon excitation in adjacent circuits, which is a key challenge in complex quantum circuits. This effect could be utilized in dense nanoscale circuits to minimize crosstalk between different components, a problem that becomes more critical as device size shrinks.

While we are most concerned with quantum applications, the spreading of quantum information via induced charge is indeed a challenge. Our work provides a potential

solution by demonstrating a mechanism to suppress this induced charge in nearby circuits. In the revised manuscript, we have added the comment on “Plasmon transistor” as a potential application of electron-channel blockade [**Correction 12**].

2. Alternative Mechanisms for Suppression

We appreciate you raising the point that the conditions for our proposed electron-channel blockade were not perfectly met in all cases (specifically for the parallel waveguide experiment where $L_p \sim 2L$). We agree that this "boundary situation" may allow for other effects to contribute to the observed suppression.

Our primary explanation for the suppression is the electron-channel blockade, which arises from the FP cavity's resonance condition not being met due to the mismatch between the wavepacket's narrow energy spectrum ($f_c, \Delta f$ in the revised manuscript) and the cavity's frequency quantization ($\Delta f, f_{FP}$ in the revised manuscript).

An alternative or complementary mechanism could be destructive quantum interference within the FP cavity. When the conditions for blockade are not met, specific frequency components of the wavepacket can establish a plasmon eigenstate within the FP cavity. However, the interaction of the wavepacket with the cavity barriers could still lead to destructive interference effects. In the intermediate regime ($L_p \sim 2L$), components of the wavepacket might interfere destructively, leading to an overall suppression of the transmitted signal without a full blockade of the electron channels. This would be a more subtle effect than a pure blockade and may account for the observed suppression even when the theoretical conditions for it are not strictly satisfied.

For a more quantitative understanding of these plasmon dynamics, numerical simulations would be helpful, although simulating plasmon dynamics within a FP cavity is a current technical challenge. We have added a comment on numerical tools as future work to support the deeper understanding of our system [**Correction 11**].

8. Last paragraph typo: “quasi-1D electron waveduides”

- We have corrected the typo [**Correction 19**].

Reviewer #3 (Remarks to the Author):

Overall Assessment

The manuscript reports time-resolved transport experiments in 2DEG waveguides that demonstrate selective propagation of 1-D plasmons by combining tunable quantum-point-contacts with a FP cavity. By injecting picosecond voltage pulses the authors show (i) constant group velocity for short pulse when many lateral channels remain active, and (ii) a strong gate-controllable velocity for long pulse when many channels are “blocked”. They attribute the observation to an electron-channel-blockade mechanism that operates when the spectral width of the pulse is narrower than the FP mode spacing. The work is technically demanding, addresses a timely question—controllable on-chip plasmonics for flying-qubit applications I recommend its publication if the following issues are fully addressed.

Major Comments

C1 Exclude or confirm dispersion-related effects

In this work, velocity tuning is interpreted purely via channel count; nevertheless plasmon dispersion (ω vs k) could also influence apparent velocities, especially for the broadband 52 ps pulse given that the short-pulse spans a broad spectrum. The authors should supply sufficient evidence that dispersion does not dominate the observed velocity trends or explicitly include its contribution, or the central conclusion is not secure.

- We thank the reviewer for raising this important point regarding plasmon dispersion. We have carefully considered this effect and believe our conclusion—that velocity tuning is primarily due to channel count—is secure.

Our understanding of the system is based on the bosonisation formalism for interacting one-dimensional electronic systems (Luttinger liquids). This theory predicts that the wavepackets we detect, which are formed by a single plasmon mode, should have an approximately **linear dispersion relation**. The theory has previously been shown to provide an excellent description of the behavior observed in similar systems by some of the authors in our prior work [G. Roussely et al., *Nat. Commun.* 9, 2811 (2018)].

The reviewer is right to be concerned about the potential for dispersion to affect the group velocity, particularly for the broadband 52 ps pulse. However, our experimental results for

the short, broadband pulse show a **constant group velocity** across a range of gate voltages. The absence of a strong dependence on gate voltage provides strong experimental evidence that the dispersion relation does not play a significant role on the observed velocity control by the QPC.

Furthermore, we do not expect the local confinement of the QPC, which extends for only a small fraction of the total waveguide length ($\sim 0.3 \mu\text{m}$ out of $50\text{--}100 \mu\text{m}$), to significantly modify the overall dispersion properties of the plasmon. The observed velocity trends are best explained by a change in the number of channels contributing to the plasmon mode, which fundamentally alters the plasmon's properties, rather than by a local dispersive effect.

We have added a more detailed explanation about the eigenmode of plasmons and its speed control. In addition, we have written that plasmons are expected to have a linear dispersion according to the bosonisation formalism in the revised manuscript [**Correction 1, 5**].

C2 Provide a plasmonic waveguide model or simulation

An analytical model or numerical simulation showing the electric field distribution along the waveguide would strengthen the argument and allow quantitative extraction of the conduction channels and FP harmonic modes.

- We appreciate the reviewer's suggestion. An analytical model or numerical simulation would indeed provide a powerful quantitative complement to our experimental findings.

However, the specific system in this paper, which involves plasmon eigenstate control using a Fabry-Pérot (FP) cavity, presents unique challenges for existing numerical methods. While some of the authors have previously performed simulations on quantum wires with uniform confinement (as referenced in the supplementary information of [G. Roussely et al., *Nat. Commun.* 9, 2811 (2018)]), these tools are not directly applicable here.

Our collaborators have attempted the simulation including the FP cavity with their numerical simulation tool, *tkwant*. They found that FP resonances in the cavity result in multiple delta-function-like contributions to the integrand from which the many-body observables are calculated. With the numerical approach currently implemented in the *tkwant* algorithm, which is based on polynomial quadratures, such highly peaked integrands cannot be handled properly, and the resulting numerical values cannot be considered reliable. We believe that a quantum tensor representation (see, for example, Fernández *et al.*, “*Learning tensor networks with tensor cross interpolation: New algorithms and libraries*”,

SciPost Phys. **18**, 104 (2025).) would allow the integral to be formulated in a way that properly accounts for highly peaked integrands arising from physical systems with resonances. However, this is beyond the scope of the present work.

We have, however, added a comment in the revised manuscript to state that the numerical simulation mentioned in the previous work is relevant to the uniform waveguide scenario (Figs. 1c,d) but is not applicable to the local confinement scheme central to this paper [**Correction 11**].

C3 Discuss why more negative gate bias induces slower plasmon

The speed of the plasmon wave in Fig. 2c are central, yet the manuscript offers only a descriptive remark. Please add related explanation to the speed variation.

- We thank the reviewer for this question. The explanation for the observed speed variation is directly related to the physical mechanism discussed in response to point 4 raised by Reviewer #2. The speed of the plasmon wavepacket is determined by the number of single-particle electronic channels that contribute to the plasmon mode.

A more negative gate voltage on the quantum point contact (QPC) effectively reduces the number of active channels. As the number of contributing channels decreases, the overall Coulomb interaction is weakened. Since the plasmon speed is directly enhanced by the strength of this interaction, a reduction in the number of channels leads to a slower plasmon mode. Our measurements of the wavepacket's group velocity directly reflect this change in the plasmon mode's speed. We have added more explanation about the plasmon mode and its speed control in the revised manuscript [**Correction1**].

C4 Define and count “electron conduction channels”

The term appears repeatedly, but there is no related discussion how N is calculated. A brief explanation of N determined by QPC would be helpful for understanding this work.

- Firstly, we would like to confirm that when we use “channel”, it means single-particle one dimensional channel. Experimentally, it is determined by the observation of the quantised conductance. For a QPC with only a few channels, we can determine the number N . The numbers appeared at the top part of Fig. 2c and Fig. 3e are determined by the observation

of the plateaux. On the other hand, for a larger number of channels quantised energy gets smaller and it becomes difficult to resolve plateaux. We added the explanation about how we determine N in the revised manuscript [**Correction 7**].

C5 Specialist vocabulary

“Fractionalisation” and “fractionalization” occur interchangeably. Use a single spelling (the literature mainly uses “fractionalization”).

- Thank you for pointing out the mixing of the spelling. We have adopted British spelling and hence use “fractionalisation” over the revised manuscript [**Correction 20**].

C6 “Plasmon eigenstate” concept not sufficiently discussed

The term appears in the title and main text, but the concept is only mentioned tangentially. It is better to add related explanations or discussion on its concept and calculation.

- Thank you for your important comment. “Plasmon eigenstate” is indeed a central concept of this paper but we did not discuss it enough. It is discussed in detail at the response to point 4 raised by Reviewer #2. We have added the explanation about the eigenstate of the wavepacket and the plasmon mode in the revised manuscript [**Correction 1**].

Minor Points

P1 Clarify notation f_c vs. Δf

In conventional RF/wave-guide literature f_c denotes the wave-guide cut-off and Δf often designates the pulse bandwidth. In this manuscript, this convention is reversed. To avoid confusion, I suggest renaming them.

- Meanwhile we use the term cut-off frequency. However, we can indeed use the “bandwidth” of the pulse instead. We use Δf as the bandwidth of the wavepacket. Then the quantised energy of the Fabry-Pérot (FP) cavity is expressed by f_{FP} in the revised manuscript [**Correction 9**].

P2 Missing label V_{w2} in Fig. 2

The value of the gate voltage applied to w_2 is absent. Please specify it either in the caption or directly in the panel.

- In the current manuscript we only mentioned that the quantum wire is kept wide. However, it is indeed important to indicate the actual gate voltages. We have added the gate voltage applied on w_1 and w_2 in the figure caption in the revised manuscript [**Correction 6**].

P3 Spelling errors

Examples: “wavepackts”, “packetes”, “quasi-1D”, “a FP cavity”, “a N-channel plasmon mode” → “wave packets”, “packets”, “quasi-1D”, “an FP cavity”, “an N-channel plasmon mode”.

- We have carefully checked the manuscript and have corrected the typos [**Correction 19**].

P4 Mixed British and American spelling

Instances such as “normalise” / “behaviour” alongside “behavior” / “color ” “funnelled”, adopt a single style.

- We have checked the manuscript and used British spelling over the manuscript [**Correction 20**].

P5 Undefined abbreviations

Define acronyms such as “FWHM” at first occurrence.

- We have defined it in the revised manuscript when it first appeared [**Correction 3**].

List of corrections

- Page and line numbers are the ones in the marked-up manuscript.
- 1. **We have modified the introduction paragraph to add the explanation about the “eigenstate” of wavepackets and speed control of the plasmon mode.**

[page2, line 37]

“In a quantum circuit composed of quasi-one-dimensional quantum wires, an eigenstate of a plasmon wave packet can be described as a charge mode of a Tomonaga-Luttinger liquid (TLL) [22]. The plasmon eigenstate in these systems is formed by all electron conduction channels below the Fermi energy, which are coupled together via Coulomb interactions. These interactions can result in non-trivial propagation of plasmons and can cause charge fractionalisation [23, 24] and spin-charge separation [25–27], as shown in quantum Hall systems with one or two conduction channels [28–30]. At zero magnetic fields, controlling the plasmon eigenstates and thus their propagation properties can be achieved by electrostatic Schottky gates [31]. By using the gates defining the quantum wire, one can control the number of available conduction channels, and thus modify on-demand the plasmon wave packet speed.”

→

“Eigenstates of a wavepacket in a quasi-one-dimensional quantum wire can be described by the bosonisation formalism [22] generalising Tomonaga-Luttinger liquid to a system containing an arbitrary number of electron conduction channels. In a quantum wire with N electron conduction channels below the Fermi energy ($2N$ including spin), the Coulomb interaction creates N charge modes and N spin modes. Here we use the term “channel” to refer to the single-particle transverse eigenstate of the Fermi liquid, and “mode” to refer to the collective excitations in the interacting system. When a wavepacket excited by a voltage pulse on an Ohmic contact, it is a charge excitation and hence only charge modes are excited. Among the N charge modes, one fast plasmon mode and $N - 1$ slower modes are formed. The plasmon mode has the highest charge-carrying capability and is the dominant eigenstate of a wavepacket excited by a short voltage pulse.

Such an interacting system has been studied in the quantum Hall regime with one or two conduction channels [23–25] and non-trivial propagation of plasmons have been revealed such as charge fractionalisation [26, 27] and spin-charge separation [28–30]. At zero magnetic fields, controlling the eigenstates of plasmon wavepackets and thus their propagation properties can be achieved by electrostatic Schottky gates [31]. By using the gates defining the quantum wire, one can control the number of available

conduction channels for the plasmon mode. Since the speed of the plasmon mode is enhanced due to the Coulomb interaction between electrons in the channels contributing to the plasmon mode compared to the Fermi velocity of Fermi liquid electrons, controlling the number of available channels can be a useful tool for controlling the plasmon speed on-demand.”

2. We have removed the misleading text.

[Page 3, line 87]

“ ~~following reference [31].~~”

3. We have defined full width at half maximum (FWHM).

[Figure 1 caption]

“**b.** Temporal shape of the voltage pulses generated by the AWG, with pulse widths defined by full width at half maximum (FWHM) ranging from 52 ps to 500 ps.”

4. We have added the explanation about V_{w1} , V_{w2} .

[Page 3, line 77]

“The waveguide length can be adjusted using two segments, labelled w1 and w2. By applying a negative gate voltage to these surface gates (V_{w1} , V_{w2}), the underlying two-dimensional electron gas (2DEG) can be depleted, allowing for the formation of either a 50 μm waveguide (w2) or a 100 μm waveguide when both segments (w1+w2) are combined.

[Figure 1 caption]

“The voltage applied on the gates w2, V_{w2} is changed from -0.6 V at the bottom to -1.2 V at the top by -0.1 V step.”

5. We have clarified that what we have measured is the group velocity and also added that plasmons have a linear dispersion according to the bosonisation formalism.

[page5, line 124]

“The group velocity of the plasmon wavepacket, v_p , is then calculated from the peak delay, t_p , and the length of the quantum wire, L_{wire} , using the relation $v_p = L_{\text{wire}}/t_p$. In the following, “speed” refers to the absolute value of the group velocity. According to the bosonisation formalism, plasmons have a linear dispersion relation and hence in principle their group velocity is equal to their phase velocity[32].”

6. We have added the gate voltage information in the caption of Figure 2, Extended

data figures 3, 4 and 5.

[Figure 2 caption]

“The gate voltage $V_{W1,W2}$ was fixed to -0.7 V.”

[Extended data figure 3 caption]

“The gate voltage V_{W2} was fixed to -0.7 V.”

[Extended data figure 4 caption]

“In this measurement, V_{QPC2} is fixed to -0.2 V and V_{W2} is fixed to -0.7 V.”

[Extended data figure 5 caption]

“The gate voltage $V_{W1,W2}$ was fixed to -0.7 V.”

- 7. We have added the explanation about how to determine the number of the electron conduction channels across QPCs in the caption of Figures 2 and 3, Extended data figures 3, 5, and 6.**

[Figure 2 caption]

“Here the number N on top of the grey shaded gate voltage indicates the number of transmitting electron channels across QPC1 at the each voltage, which is determined by the observation of the quantised conductance.”

[Figure 3 caption]

“Here the number N on top of the grey shaded gate voltage indicates the number of transmitting electron channels across QPC1, which is determined by the observation of the quantised conductance.”

[Extended data figure 3 caption]

“Here the number N on top of the grey shaded gate voltage indicates the number of transmitting electron channels across QPC2, which is determined by the observation of the quantised conductance.”

[Extended data figure 5 caption]

“Here the number N on top of the grey shaded gate voltage indicates the number of transmitting electron channels across QPC1, which is determined by the observation of the quantised conductance.”

[Extended data figure 6 caption]

“The number N on top of the grey shaded gate voltage indicates the number of transmitting electron channels across g_{QPC} , which is determined by the observation of the quantised conductance.”

- 8. We have added the analogy between “charge fractionalisation” and “mode-matching” in optics.**

[Page 6, line158]

“This result indicates that the plasmon **wavepacket** is being redistributed to the eigenstates formed by all the conduction channels in the waveguide after passing through QPC1 locally limiting the number of the transmitting conduction channels, which bares similarities to mode-matching in optics, when coupling multimode waveguide/cavity with different eigenmodes.”

- 9. We have changed the cut-off frequency $f_c \rightarrow$ the bandwidth Δf and the quantised energy of the FP cavity from $\Delta f \rightarrow f_{\text{FP}}$ in section 3 and in Figure 3.**

- 10. We have modified the explanation about Figure 3e in the caption to explicitly show which data is new data and which data is the same as in Figure 2c.**

[Figure 3 caption]

“e. Comparison of plasmon speed as a function of V_{QPC1} with and without the FP cavity for the wavepackets excited by 500 ps pulse. The data without the FP cavity (new data) and the data with the FP cavity (from Fig. 2c) is provided for direct comparison.”

- 11. We have added the comments on numerical simulations to understand the observed behaviour.**

[Page 10, line 287]

“For a more quantitative understanding of the observed behaviour numerical simulations would be useful. In a previous work, controlling the plasmon eigenstate by modifying the width of the whole waveguide, was well understood with parameter-free numerical simulations [31]. However, direct application of the same numerical tool is not possible here due to the complex resonance behaviour of the FP cavity. The development of new numerical tools allowing for the simulation of the system with the FP cavity is desirable and will help us gain deeper understanding of the system.”

- 12. We have added the comment on classical applications of electron-channel blockade in the conclusion paragraph.**

[Page 10, line 314]

“The capability to switch on or off the induced charge or more widely the propagation of a specific plasmon eigenstate may be of interest in applications for classical plasmonic circuits, like a plasmon transistor.”

- 13. We have added the discussion about the uncertainty of our measurement to calculate the plasmon velocity in Methods.**

[Page 11, line 339]

“This time delay, t_{2D} , is expressed by $t_{2D} = (L_0 + 100 \mu\text{m})/v_{2D} + \delta_{\text{line}}$, where L_0 is the distance between the injection contact, O_i and QPC1, v_{2D} is 2D plasmon velocity, and δ_{line} is the time delay between the RF lines connected to O_i and QPC3. For the time-resolved measurement the delay at the peak, t_{peak} is $t_{\text{peak}} = (L_0 + 100 \mu\text{m} - L_{\text{wire}})/v_{2D} + L_{\text{wire}}/v_p + \delta_{\text{line}}$, where L_{wire} is the length of the quantum wire used as a waveguide.

Therefore, $t_p = L_{\text{wire}}/v_p$ is equal to $(t_{\text{peak}} - (t_{2D} - L_{\text{wire}}/v_{2D}))$. We consider that $v_{2D} = 1.0 \pm 0.2 \times 10^7$ m/s. As a result, we use 102 ps (107 ps) as a zero time delay for the plasmon wavepacket time-resolved measurement data with the uncertainty of $\delta_{\text{calibration}} = 10$ ps (9 ps) for the data obtained in 100 μm (50 μm) quantum wire. The same calibration is performed for the device in Fig. 4. Then the plasmon velocity is calculated by $v_p = L_{\text{wire}}/t_p$. Here the total uncertainty of t_p , $\delta_{\text{tot}} = \delta_{\text{calibration}} + \delta_{\text{fitting}}$ is used to calculate the error bar of the measurement.”

- 14. We have added the discussion to compare the plasmon speed control by QPCs for 50 μm quantum wire and 100 μm quantum wire in the caption of Extended data figure 3.**

[Extended data figure 3 caption]

“The data in **a** is in contrast to the data in Fig. 2a in 100 μm quantum wire. For this 50 μm quantum wire case, the wavepacket length is neither much smaller nor larger than double the cavity length ($L_p \sim 2L$), representing an intermediate, or “boundary”, situation. The result is a shift in the peak delay as a function of QPC2 voltage, but the main peak is followed by several sub-peaks. This suggests that the wavepacket is not in a well-defined eigenstate and spreads into several different charge modes after passing through QPC2.”

- 15. We have added a “Data availability” statement.**

Data availability

The datasets generated and analysed during the current study are available in the Zenodo repository, [<https://doi.org/10.5281/zenodo.16879800>]. All other relevant data are available from the corresponding author upon request.

- 16. We have corrected the error bar of the figures (Figure 1d, Figure 2c, Figure 3a, 3e, Extended data figure 3c, Extended data figure 5c, and Extended data figure 6d).**

- 17. We have cleaned up and reanalysed the data to upload them to the repository for data availability. While reanalysing the data, some numbers appeared in the manuscript has been slightly modified (e.g. 2D plasmon delay time 111 ± 6 ps \rightarrow 112 ± 7 ps).**

- 18. “Wave packet”, “wave packets” have been modified to “wavepacket”, “wavepackets” throughout the manuscript.**

- 19. We have corrected several typos in the manuscript.**

- 20. We have used British spelling in the revised manuscript.**

- There are no questions or comments from **Reviewer #1** and **Reviewer #2**.

Reviewer #3 (Remarks to the Author):

I thank the authors for the detailed responses provided to the reviewer's previous comments. While I am satisfied with the clarifications on most of the concerns raised, a further point has emerged from the authors' response.

I would like to inquire whether the “plasmon eigenstate” described in the manuscript can be understood in a similar fashion to the SPP modes found in IMI (insulator-metal-insulator) structures, where SPPs split into high-energy and low-energy branches. The splitting width is modulated by the core thickness, becoming larger as the thickness decreases, resulting in a larger wave number modulation. The bosonization formalism presented in this paper appears to exhibit a similar phenomenon, where the plasmon velocity and mode characteristics are modulated by the transverse channel configuration, analogous to how the SPP splitting in IMI systems is controlled by the thickness of the intermediate layer.

The reviewer seeks confirmation whether this analogy holds. Addressing this analogy could potentially broaden the article's understanding and impact within the classical surface plasmonics community.

- We thank the reviewer for the thoughtful analogy between our plasmon eigenstates and the SPP modes in IMI structures. While we agree that there is a similarity in that, in the both cases, the mode characteristics are modulated by the width of the quantum wire (or the middle metal layer), we would like to clarify several important distinctions.
 - **Dispersion Relation:** The plasmon mode in a quasi-1D quantum wire exhibits a linear dispersion, in contrast to the more complex, quadratic-like dispersion of SPPs in IMI structures.
 - **Mode Multiplicity:** A simple IMI structure supports only two SPP branches (high- and low-energy). In our system, with N non-interacting conduction channels (or $2N$ including spin), we obtain N spin modes and N charge modes. Among the charge modes, one special mode – the plasmon mode – dominates charge transport. When a

wavepacket is excited via a voltage pulse at an Ohmic contact, it primarily couples to this plasmon mode.

- **Opposite Tendency in Splitting:** In IMI structures, the energy splitting increases as the middle metal layer becomes thinner. In contrast, in our system, a wider quantum wire supports more conduction channels, leading to stronger Coulomb interactions and a faster plasmon mode. The other charge modes remain near charge neutrality and propagate close to the Fermi velocity, resulting in a larger energy separation for wider wires – an opposite trend.
- **Alternative Analogy:** The energy splitting between non-interacting conduction channels indeed increases as the wire becomes narrower (See Supplementary Figure 6 of Ref. 31), which may offer a closer analogy to IMI-SPP splitting. However, our focus is on the plasmon mode arising from Coulomb interactions between these channels, which is fundamentally different from the electromagnetic origin of SPPs.

We include a note in Supplementary Information to acknowledge this analogy while clarifying the distinctions, in order to avoid potential confusion and to help readers appreciate the unique nature of our system.

List of modifications

- We added the reference [new reference 32] to Supplementary Note about the analogy between the plasmon mode and IMI-SPP splitting.
- We added Supplementary Note about the analogy between the plasmon mode and IMI-SPP splitting in Supplementary Information.

Reviewer Report

Eigenstate Control of Plasmon Wave-Packets with Electron-Channel Blockade

Overall Assessment

The manuscript reports time-resolved transport experiments in 2DEG waveguides that demonstrate selective propagation of 1-D plasmons by combining tunable quantum-point-contacts with a FP cavity. By injecting picosecond voltage pulses the authors show (i) constant group velocity for short pulse when many lateral channels remain active, and (ii) a strong gate-controllable velocity for long pulse when many channels are “blocked”. They attribute the observation to an electron-channel-blockade mechanism that operates when the spectral width of the pulse is narrower than the FP mode spacing. The work is technically demanding, addresses a timely question—controllable on-chip plasmonics for flying-qubit applications

I recommend its publication if the following issues are fully addressed.

Major Comments

C1 Exclude or confirm dispersion-related effects

In this work, velocity tuning is interpreted purely via channel count; nevertheless plasmon dispersion (ω vs k) could also influence apparent velocities, especially for the broadband 52 ps pulse given that the short-pulse spans a broad spectrum. The authors should supply sufficient evidence that dispersion does not dominate the observed velocity trends or explicitly include its contribution, or the central conclusion is not secure.

C2 Provide a plasmonic waveguide model or simulation

An analytical model or numerical simulation showing the electric field distribution along the waveguide would strengthen the argument and allow quantitative extraction of the conduction channels and FP harmonic modes.

C3 Discuss why *more negative gate bias induces slower plasmon*

The speed of the plasmon wave in Fig. 2c are central, yet the manuscript offers only a descriptive remark. Please add related explanation to the speed variation.

C4 Define and count “electron conduction channels”

The term appears repeatedly, but there is no related discussion how N is calculated. A brief explanation of N determined by QPC would be helpful for understanding this work.

C5 Specialist vocabulary

“Fractionalisation” and “fractionalization” occur interchangeably. Use a single spelling (the literature mainly uses “fractionalization”).

C6 “Plasmon eigenstate” concept not sufficiently discussed

The term appears in the title and main text, but the concept is only mentioned tangentially. It is better to add related explanations or discussion on its concept and calculation.

Minor Points

P1 Clarify notation f_c vs. Δf

In conventional RF/wave-guide literature f_c denotes the wave-guide cut-off and Δf often designates the pulse bandwidth. In this manuscript, this convention is reversed. To avoid confusion, I suggest renaming them.

P2 Missing label V_{w2} in Fig. 2

The value of the gate voltage applied to w2 is absent. Please specify it either in the caption or directly in the panel.

P3 Spelling errors

Examples: “*wavepackts*”, “*packetes*”, “*quasi-1D*”, “a FP cavity”, “a N-channel plasmon mode” → “*wave packets*”, “*packets*”, “*quasi-1D*”, “an FP cavity”, “an N-channel plasmon mode”.

P4 Mixed British and American spelling

Instances such as “*normalise*” / “*behaviour*” alongside “*behavior*” / “*color*” “*funnelled*”, adopt a single style.

P5 Undefined abbreviations

Define acronyms such as “FWHM” at first occurrence.

Recommendation

Publish after major revision.